# VLM-Guided Noisy Label Detection
# for Structured Network Traffic in Low-Resource IDS

**Yuha Jeon** [* 1]  **Hyoung-Kyu Song** [2]  **Myung-Sun Baek** [1]

## Abstract

Accurate Intrusion Detection Systems (IDS) rely on high-quality labeled network traffic data, yet real-world deployments suffer from scarce annotations and label noise introduced during automated or crowdsourced labeling pipelines. We present VLM-IDS, a multimodal noisy label detection pipeline combining Co-Teaching-based filtering, Adaptive Cluster Density (ACD) sub-clustering, and a fine-tuned Vision Language Model (VLM). Our approach converts per-feature flow statistics into three-channel Shannon Entropy density maps, enabling a Qwen2-VL-2B model to localize high-risk feature bin regions and guide selective augmentation and density-aware feature injection into a Random Forest classifier. Experiments on Do-HBrw2020 and Gotham2025 under symmetric noise rates of 20–45% with $N = 250$–$1000$ show consistent improvements over a raw RF baseline and the prior method RAPIER, with up to 13.4 percentage-point F1 improvement on Do-HBrw2020 at $p = 0.40$.

## 1. Introduction

The rapid growth of network traffic—with diverse protocols spanning encrypted HTTPS, IoT device communication, and DNS tunneling—has pushed network intrusion detection systems (IDS) toward flow-level statistical features such as packet lengths, inter-arrival times, and TCP flag sequences, as payload-level inspection is often infeasible or insufficient. While machine learning classifiers trained on these features achieve high accuracy under clean conditions, two practical challenges severely limit real-world deployment.

First, label noise is pervasive. Crowdsourced annotation platforms introduce human error; automated labeling tools misclassify edge cases; adversarial actors deliberately inject mislabeled samples to degrade classifier quality. Under symmetric noise rates above 30%, standard classifiers experience catastrophic performance degradation—even robust ensembles such as Random Forest and XGBoost lose 10–20 percentage points in F1, as our experiments confirm. Notably, recent re-annotation studies on CICIDS2017 (Engelen et al., 2021) found mislabeling rates up to 41.7% in specific traffic subsets, validating the 10–40% experimental range we evaluate as operationally realistic.

Second, labeled data is scarce. Capturing, labeling, and auditing network traffic is expensive and time-consuming. Many operational IDS deployments work with only a few hundred to a few thousand labeled flows—a regime where noise has an outsized impact and data augmentation is critical.

Existing noisy label methods were designed for image or text data and do not leverage the structured, tabular nature of network traffic. The prior IDS-specific method RAPIER (Qing et al., 2023) addresses low-quality data through generative correction but relies on stable noise distributions and suffers high variance under symmetric noise, falling below F1= 0.30 at noise rates above 40% in our evaluation.

We propose VLM-IDS, which reframes noisy label detection as a multimodal structured learning problem. The core insight is that tabular flow statistics can be projected into semantically meaningful two-dimensional density images—where the row axis encodes features, the column axis encodes histogram bins, and the Shannon Entropy channel highlights regions where benign and malicious flows co-occur. A fine-tuned Vision-Language Model (VLM) then identifies these high-risk regions, producing both a natural-language explanation for human analysts and a structured JSON output for downstream processing. This tabular-to-image bridge enables foundation model capabilities to benefit structured time-series data, directly addressing the multimodal structured learning focus of the ICML 2026 FMSD workshop.

Our contributions are: (1) Shannon Entropy density maps

[1]Department of Artificial Intelligence and Information Technology, Sejong University, Seoul, South Korea [2]Department of AI Convergence Electronics Engineering, Sejong University, Seoul, South Korea. Correspondence to: Myung-Sun Baek <msbaek@sejong.ac.kr>.

*Proceedings of the $2^{nd}$ ICML Workshop on Foundation Models for Structured Data*, Seoul, South Korea. 2026. Copyright 2026 by the author(s).

that make noisy label regions visually salient for VLM consumption; (2) an overlap-adaptive routing strategy that applies VLM guidance only when class confusion warrants it; (3) density-aware feature injection that propagates VLM risk signals directly into the downstream classifier; and (4) a comprehensive empirical study across two datasets, three training sizes, and four noise rates with ten random seeds.

## 2. Related Work

### 2.1. Noisy Label Learning

The memorization effect of deep networks—they fit clean samples before noisy ones—motivates a family of sample-selection methods. Co-Teaching (Han et al., 2018) trains two networks in parallel, each selecting small-loss samples for the other to update on. SELF (Nguyen et al., 2020) uses self-supervised pretraining to distinguish clean from noisy samples. DivideMix (Li et al., 2020) models per-sample loss as a Gaussian mixture to obtain soft clean/noisy assignments. These methods are designed for deep neural networks on image data; adapting them to the tabular, low-data IDS setting requires architectural changes we avoid by using a Random Forest classifier with an upstream filtering pipeline.

Generalized Cross Entropy (GCE) (Zhang & Sabuncu, 2018) provides a noise-robust loss function applicable to any differentiable model, interpolating between cross-entropy (noise-sensitive) and mean absolute error (noise-robust). Confident Learning (CL) (Northcutt et al., 2021) estimates the joint distribution of noisy and true labels from out-of-fold predicted probabilities, enabling model-agnostic identification of label issues. We include CL as a baseline in our evaluation; results are reported in Appendix D.

### 2.2. Network Traffic Classification and IDS

Flow-level machine learning for IDS has been studied extensively since the CIC datasets (Sharafaldin et al., 2018). Rather than relying on payload content—which may be encrypted, obfuscated, or simply unavailable in many deployment scenarios—these methods operate on sequence-level statistics such as packet length distributions and inter-arrival timing patterns, which form the 50-dimensional feature space in our work. RAPIER (Qing et al., 2023) is the closest prior work: it uses a GRU autoencoder to extract latent features, MADE-based density estimation to identify clean samples, and a GAN to generate synthetic training data correcting for noise. RAPIER performs well under asymmetric or low noise but, as our experiments show, degrades sharply under symmetric noise above 40%.

### 2.3. Foundation Models for Structured Data

TabPFN (Hollmann et al., 2022) demonstrates that transformer architectures pretrained on synthetic tabular tasks can match or exceed gradient-boosted trees on small datasets in a zero-shot manner. TabICL and TabDPT extend this to in-context learning on tabular classification. These methods do not address noisy labels—our work is complementary, focusing on data cleaning rather than model pretraining. On the vision side, Qwen2-VL (Wang et al., 2024) achieves strong performance on spatial reasoning tasks, which we leverage to localize high-overlap density regions. VLM-IDS is the first work to use a VLM as a noisy label detector for tabular time-series data.

## 3. Methodology

### 3.1. Stage 1: CT Filter

We apply Co-Teaching using XGBoost base learners with progressive keep-rate scheduling from 70% to 85% over five rounds. In each round, both learners independently select their small-loss subsets; only their intersection is retained as $X_{ct}, y_{ct}$. This progressive schedule prevents aggressive early rejection of borderline-clean samples while achieving strong noise reduction at convergence.

### 3.2. Stage 2: Shannon Entropy Density Map

From $X_{ct}$, we construct a $50 \times 50$ density map via per-feature kernel density estimation, where the row axis encodes feature dimensions and the column axis encodes histogram bins. Three channels encode complementary information. Channel 1 (R) encodes $\log(1 + \cdot)$-transformed benign density; Channel 2 (G) encodes $\log(1 + \cdot)$-transformed malicious sample density; Channel 3 (B) encodes the Shannon Entropy $H(p_{be}, p_{ma}) = -(p_{be} \log_2 p_{be} + p_{ma} \log_2 p_{ma})$ per cell, where $p_{be}$ and $p_{ma}$ are the normalized benign and malicious densities. Cells where both classes co-occur receive high entropy and appear bright in Channel 3, making noisy overlap regions visually salient for the VLM. The map is resized to $224 \times 224$ RGB for VLM input.

### 3.3. Stage 3: VLM Analysis and Overlap-Adaptive Routing

The density image is passed to a fine-tuned Qwen2-VL model (LoRA $r = 32$, $\alpha = 64$; vision encoder fully fine-tuned). The model produces a single response containing two components: (1) a natural-language explanation of the identified risk regions for human interpretability, and (2) a structured RESULT JSON from which risky features and risky bins are parsed. Fine-tuning used 71,280 QA pairs derived from ground-truth mislabeling positions over 13

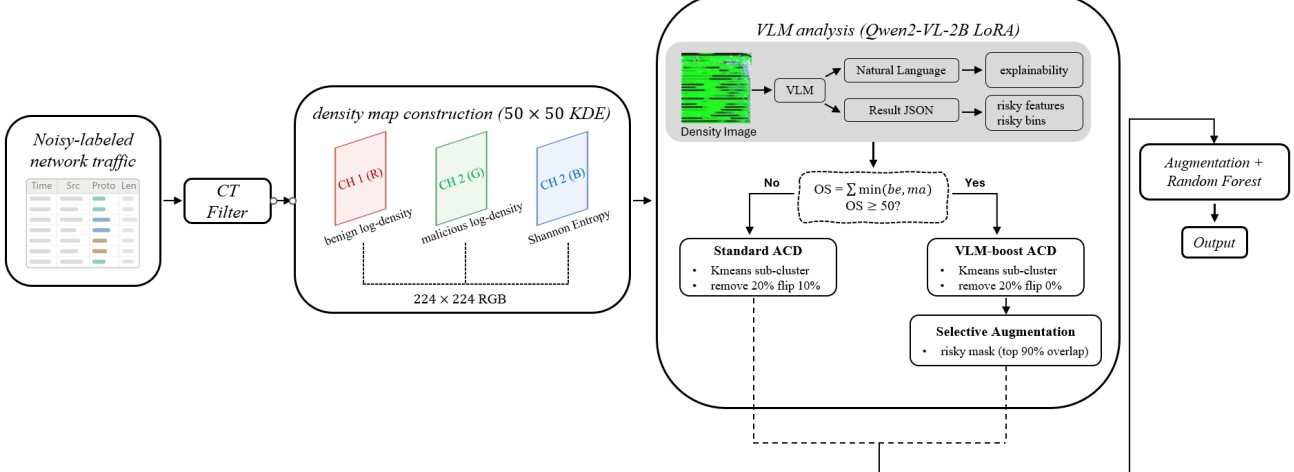

*Figure 1.* Overview of the VLM-IDS pipeline. Noisy-labeled network traffic is first filtered by a Co-Teaching-based CT filter, then projected into a three-channel Shannon Entropy density map. The density image is analyzed by a fine-tuned *Qwen2-VL-2B* model, producing a natural-language explanation and a structured RESULT JSON containing risky feature and bin indices. An overlap score $OS = \sum_{f,b} \min(d^{be}_{f,b}, d^{ma}_{f,b})$ routes each sample to either the Standard ACD path or the VLM-boost ACD path with Selective Augmentation on high-risk regions. The cleaned and augmented training data is used to fit a Random Forest classifier.

epochs.

The Overlap Score is defined as:

$$OS = \sum_{f,b} \min\left(d^{be}_{f,b}, d^{ma}_{f,b}\right) \qquad (1)$$

where $d^{be}_{f,b}$ and $d^{ma}_{f,b}$ denote the benign and malicious density at feature $f$, bin $b$. This determines the routing strategy:

$$\text{Routing} = \begin{cases} \text{VLM-guided augmentation} & \text{if } OS \geq 50 \\ \text{standard ACD path} & \text{if } OS < 50 \end{cases}$$

DoH2020 yields $OS = 47$–$64$ (high class overlap) and Gotham2025 low-noise yields $OS = 30$–$32$ (high separability).

### 3.4. Stage 4: ACD Sub-clustering

Adaptive Cluster Density applies KMeans sub-clustering ($k \leq 3$) within each class. Impure sub-clusters are flagged for removal via per-sample noisy scores; ambiguous sub-clusters are designated U-clusters (zero-day candidates) and passed through unmodified. When the ACD clean ratio falls below 0.75, the CT output is used directly to prevent over-removal.

### 3.5. Stage 5: Selective Augmentation and Feature Injection

Under the VLM-guided path, samples falling in the top-90th-percentile overlap of VLM-identified feature-bin cells

receive additional Gaussian noise and Mixup augmentation (Selective Augmentation). To further propagate VLM risk signals into the classifier, we introduce two feature injection strategies. First, the per-sample noisy score computed by ACD is injected as an additional Random Forest input feature, giving the classifier direct access to label reliability information. Second, per-feature statistics extracted from the training density map (mean benign density, mean malicious density, and Shannon Entropy for each of the 50 features, totaling 150 additional features) are appended to the classifier input. Both methods apply only under the VLM-guided path. For inference, the density statistics computed from the training set are reused as fixed additional features appended to each test sample, and the noisy score is set to zero since test labels are assumed clean.

## 4. Experiments

### 4.1. Setup

We evaluate DoH2020 (DNS-over-HTTPS tunneling, high class overlap, $OS = 47$–$64$) and Gotham2025 (IoT traffic, high natural separability, $OS = 30$–$55$), both with 50-dimensional packet-length sequence features. Training sets of $N \in \{250, 500, 1000\}$ balanced samples; test sets of 2,000 balanced samples. Symmetric noise at $p \in \{0.20, 0.30, 0.40, 0.45\}$; all results averaged over 10 random seeds.

*Table 1.* Macro F1 on DoH2020 (10-seed average). $N = 250$ RAPIER results from 3 seeds.

| N | p | Baseline | XGBoost | RAPIER | VLM-IDS |
|---|---|---|---|---|---|
| 250 | 0.20 | 0.8985 | 0.8597 | 0.6871 | **0.9221** |
| | 0.30 | 0.8055 | 0.7580 | 0.5944 | **0.8624** |
| | 0.40 | 0.6667 | 0.6403 | 0.4928 | **0.7332** |
| | 0.45 | 0.5955 | 0.5720 | 0.2997 | **0.6908** |
| 500 | 0.20 | 0.9153 | 0.8785 | 0.7452 | **0.9399** |
| | 0.30 | 0.8331 | 0.7856 | 0.6490 | **0.9023** |
| | 0.40 | 0.6862 | 0.6493 | 0.2604 | **0.9023** |
| | 0.45 | 0.6047 | 0.5813 | 0.2826 | **0.6938** |
| 1000 | 0.20 | 0.9289 | 0.8983 | 0.6526 | **0.9567** |
| | 0.30 | 0.8485 | 0.8077 | 0.2673 | **0.9195** |
| | 0.40 | 0.7028 | 0.6671 | 0.2378 | **0.7744** |
| | 0.45 | 0.6100 | 0.5874 | 0.3292 | **0.6655** |

*Table 2.* Macro F1 on Gotham2025 (10-seed average). $N = 250$ RAPIER results from 3 seeds.

| N | p | Baseline | XGBoost | RAPIER | VLM-IDS |
|---|---|---|---|---|---|
| 250 | 0.20 | 0.9438 | 0.9510 | 0.3333 | **0.9581** |
| | 0.30 | 0.8833 | 0.8875 | 0.3333 | **0.9332** |
| | 0.40 | 0.7505 | 0.7472 | 0.3349 | **0.7874** |
| | 0.45 | 0.6429 | 0.6250 | 0.2878 | **0.7002** |
| 500 | 0.20 | 0.9603 | 0.9608 | 0.5431 | **0.9784** |
| | 0.30 | **0.9322** | 0.9253 | 0.5365 | 0.9078 |
| | 0.40 | 0.8190 | 0.8106 | 0.2864 | **0.8467** |
| | 0.45 | 0.7048 | 0.7086 | 0.2924 | **0.7597** |
| 1000 | 0.20 | 0.9712 | 0.9705 | 0.5367 | **0.9783** |
| | 0.30 | 0.9438 | **0.9464** | 0.3032 | 0.9433 |
| | 0.40 | 0.8333 | 0.8364 | 0.3333 | **0.8642** |
| | 0.45 | 0.6985 | 0.7021 | 0.2342 | **0.7375** |

## 4.2. Baselines

We compare against three baselines. Raw RF trains a Random Forest directly on raw, unscaled features with noisy labels – no preprocessing —providing the lower bound. XGBoost (no preprocessing) uses gradient-boosted trees on raw noisy features, representing the strongest classical approach without label correction. RAPIER applies the prior-art pipeline (AE+MADE+GAN) to the same 50-dimensional feature data. Confident Learning (CL) (Northcutt et al., 2021) applies cross-validation predicted probabilities to estimate and remove label issues; full CL results are reported in Appendix D.

## 5. Results

VLM-IDS outperforms all baselines in 12/12 DoH2020 conditions and 10/12 Gotham2025 conditions. RAPIER degrades sharply at $p \geq 0.40$, falling below F1$= 0.30$ in both datasets—VLM-IDS maintains consistent performance across all noise rates. The strongest classical baseline, XG-Boost without preprocessing, is outperformed by VLM-IDS by up to 16.5 percentage points in DoH2020 ($N = 500$, $p = 0.40$), confirming the need for noise-aware preprocessing. The four Gotham2025 exceptions occur at $p = 0.30$ where high natural separability reduces the benefit of the VLM-guided intervention.

## 6. Discussion

### 6.1. Interpretability

Beyond classification performance, VLM-IDS produces a natural-language explanation per inference identifying which flow features contribute to label uncertainty. This supports human auditability in critical infrastructure deployments where operators need to validate model behavior.

### 6.2. Limitations

Two limitations affect current performance. First, ACD over-removal at low noise rates: when $p \leq 0.20$, Co-Teaching filtering already provides sufficient noise correction, and the subsequent ACD stage occasionally removes clean borderline samples, contributing to the four Gotham2025 exceptions. A calibrated ACD threshold that accounts for the CT filter's prior noise reduction would address this. Second, density feature dimensionality: the 150 additional features occasionally act as noise for datasets with high natural separability, where the original 50 features provide strong discriminative signal. Adaptive feature selection conditioned on the overlap score would mitigate this issue.

### 6.3. Future Work

We identify three directions for improvement. First, upgrading to a larger VLM is expected to improve risky feature localization—our 2B model shows repetitive output patterns suggesting limited spatial reasoning capacity. Second, a joint CT-ACD calibration procedure would prevent unnecessary clean sample loss in low-noise settings, addressing the ACD over-removal issue. Third, richer fine-tuning data with more diverse seed and noise-rate combinations, combined with VLM answer verification against ground-truth overlap maps, would improve alignment between VLM predictions and actual mislabeling patterns.

## 7. Conclusion

VLM-IDS demonstrates that tabular-to-image projection enables foundation model spatial reasoning to benefit structured time-series noisy label detection. Shannon Entropy density maps provide both a VLM-interpretable signal and a human-readable explanation of label uncertainty—supporting explainability requirements in IDS deployments. Future work will explore 7B-parameter VLMs, adaptive

ACD thresholds, and online retraining distillation.

## Acknowledgements

This work was supported by the Defense ICT Innovation Technology Program of the Institute of Information & Communications Technology Planning & Evaluation (IITP), funded by the Korean government (Ministry of National Defense) in 2025 (No. RS-2025-02363049, Development of dynamic trust connection and intelligent management technology for hybrid multi-layered network).

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

## A. Classical Baselines (No Preprocessing)

*Table 3.* Classical baselines on DoH2020.

| N | p | XGBoost | MLP | ExtraTrees |
|---|---|---|---|---|
| 250 | 0.20 | 0.8597 | 0.7533 | 0.8597 |
| | 0.30 | 0.7580 | 0.6810 | 0.7580 |
| | 0.40 | 0.6403 | 0.6074 | 0.6403 |
| | 0.45 | 0.5720 | 0.5386 | 0.5720 |
| 500 | 0.20 | 0.8785 | 0.6987 | 0.8785 |
| | 0.30 | 0.7856 | 0.6475 | 0.7856 |
| | 0.40 | 0.6493 | 0.6118 | 0.6493 |
| | 0.45 | 0.5813 | 0.5186 | 0.5813 |
| 1000 | 0.20 | 0.8983 | 0.7841 | 0.8983 |
| | 0.30 | 0.8077 | 0.6498 | 0.8077 |
| | 0.40 | 0.6671 | 0.5817 | 0.6671 |
| | 0.45 | 0.5874 | 0.5226 | 0.5874 |

All baselines trained on raw features with noisy labels. No preprocessing applied. 10-seed average.

*Table 4.* Classical baselines on Gotham2025.

| N | p | XGBoost | MLP | ExtraTrees |
|---|---|---|---|---|
| 250 | 0.20 | 0.9510 | 0.9061 | 0.9290 |
| | 0.30 | 0.8875 | 0.8072 | 0.8653 |
| | 0.40 | 0.7472 | 0.6480 | 0.7375 |
| | 0.45 | 0.6250 | 0.5437 | 0.6160 |
| 500 | 0.20 | 0.9608 | 0.9246 | 0.9527 |
| | 0.30 | 0.9253 | 0.8919 | 0.9110 |
| | 0.40 | 0.8106 | 0.6829 | 0.8038 |
| | 0.45 | 0.7086 | 0.5726 | 0.6929 |
| 1000 | 0.20 | 0.9705 | 0.9195 | 0.9681 |
| | 0.30 | 0.9464 | 0.8844 | 0.9359 |
| | 0.40 | 0.8364 | 0.7009 | 0.8213 |
| | 0.45 | 0.7021 | 0.5789 | 0.7007 |

All baselines trained on raw features with noisy labels. No preprocessing applied. 10-seed average.

## B. Implementation Details

**VLM Fine-tuning.** We use *Qwen2-VL-2B-Instruct* as the base model. LoRA is configured with rank $r = 32$, scaling factor $\alpha = 64$, and dropout 0.05, applied to the language model layers. The vision encoder is fully fine-tuned with a learning rate of $2 \times 10^{-5}$, which is $5\times$ higher than that of the LoRA layers. Training is conducted for 13 epochs with a batch size of 4 and gradient accumulation over 4 steps. We use the AdamW optimizer with a warmup ratio of 0.1. The dataset consists of 71,280 QA pairs (24 QA types per image $\times$ 2,970 unique images per dataset configuration). The best checkpoint is selected based on validation loss on a 10% held-out split (validation loss: 1.0126).

**Classifier.** We use a Random Forest classifier with 500 estimators, no depth limit, and `random_state=seed`.

Data augmentation includes Gaussian noise ($\sigma = 0.02$) and Mixup ($\alpha = 0.2$), resulting in approximately $2.3\times$ dataset expansion. The Co-Teaching (CT) filter is applied for 5 rounds with a keep-percentage schedule of $70\% \rightarrow 74\% \rightarrow 78\% \rightarrow 81\% \rightarrow 85\%$. XGBoost base learners are used with 100 estimators.

**ACD Parameters.** We set `max_k = 3`, `minor_thresh = 0.15`, `pure_thresh = 0.80`, and `cons_thresh = 0.70`. The number of nearest neighbors is defined as $k_{\mathrm{knn}} = \min(15, N/10)$. We remove the top 20% of samples (`top_remove_pct = 20%`) and flip the top 10% (`top_flip_pct = 10%`) for the Standard pipeline, while no flipping is applied in the VLM-based pipeline. The clean ratio threshold is set to 0.75.

**Overlap Threshold.** The overlap score $OS$ is defined in Equation (1). We use a threshold of 50. For DoH2020, the typical range is 47–64, where `use_vlm=True` is selected for most conditions. For Gotham2025, when $p \leq 0.30$, the range is 30–40 (`use_vlm=False`), and when $p \geq 0.40$, the range is 35–55 (mixed behavior).

**Data Independence.** Fine-tuning images and test-set density maps are generated from independent random seeds, ensuring no overlap between the fine-tuning corpus and evaluation conditions.

## C. Dataset Details

**DoH2020.** This dataset is collected from benign browser traffic and DNS-over-HTTPS tunneling tools, including *dns2tcp*, *dnscat2*, and *iodine*. Each flow is represented by the first 50 packet lengths, zero-padded if necessary. The full dataset is approximately balanced between benign and malicious samples; for each experiment, we subsample $N/2$ instances per class. The test set consists of 1,000 benign and 1,000 malicious samples drawn from a held-out portion of the original dataset.

**Gotham2025.** This dataset contains IoT device traffic (benign: *Philips Hue*, *Nest Cam*, *Samsung SmartThings*) and network flooding attacks (TCP SYN, UDP flood, HTTP flood). Features are constructed in the same way as DoH2020, using 50-dimensional packet-length sequences. Class balance is maintained by equal subsampling per experiment. The test set includes $\min(1{,}000, \text{ available})$ samples per class from a held-out split.

## D. Confident Learning Baseline

Confident Learning (CL) (Northcutt et al., 2021) estimates the joint distribution of observed noisy labels and latent true labels from out-of-fold predicted probabilities, then

removes samples most likely to be mislabeled. We apply CL using a 5-fold cross-validated XGBoost classifier (100 estimators) on the same standardized features used by VLM-IDS, followed by the same Random Forest classifier and augmentation pipeline.

CL performs competitively at low noise rates ($p \leq 0.30$), confirming it as a strong model-agnostic baseline. At high noise rates on DoH2020—the high class-overlap dataset—VLM-IDS outperforms CL in most conditions, with the largest gap at $N = 500$, $p = 0.40$ (VLM-IDS 0.9023 vs. CL 0.7924, +10.9 pp). This advantage is consistent with the density map's ability to localize overlap regions that per-sample confidence scores cannot distinguish under high class confusion. On Gotham2025, where natural class separability is high, CL is competitive or superior, reflecting that confidence-based filtering suffices when features are inherently discriminative.

*Table 5.* Confident Learning vs. VLM-IDS on DoH2020 (10-seed average Macro F1).

| N | p | Baseline | CL | VLM-IDS |
|---|---|---|---|---|
| 250 | 0.20 | 0.8985 | **0.9478** | 0.9221 |
| | 0.30 | 0.8055 | **0.8672** | 0.8623 |
| | 0.40 | 0.6667 | **0.7392** | 0.7332 |
| | 0.45 | 0.5955 | 0.6395 | **0.6908** |
| 500 | 0.20 | 0.9153 | **0.9675** | 0.9399 |
| | 0.30 | 0.8331 | **0.9286** | 0.9023 |
| | 0.40 | 0.6862 | 0.7924 | **0.9023** |
| | 0.45 | 0.6047 | 0.6853 | **0.6938** |
| 1000 | 0.20 | 0.9289 | **0.9801** | 0.9567 |
| | 0.30 | 0.8485 | **0.9468** | 0.9195 |
| | 0.40 | 0.7028 | **0.8093** | 0.7986 |
| | 0.45 | 0.6100 | 0.6839 | **0.7002** |

*Table 6.* Confident Learning vs. VLM-IDS on Gotham2025 (10-seed average Macro F1).

| N | p | Baseline | CL | VLM-IDS |
|---|---|---|---|---|
| 250 | 0.20 | 0.9438 | **0.9733** | 0.9546 |
| | 0.30 | 0.8833 | **0.9305** | 0.9266 |
| | 0.40 | 0.7505 | **0.8202** | 0.7844 |
| | 0.45 | 0.6429 | 0.6646 | **0.7002** |
| 500 | 0.20 | 0.9603 | **0.9852** | 0.9776 |
| | 0.30 | 0.9322 | **0.9586** | 0.8858 |
| | 0.40 | 0.8190 | **0.8499** | 0.8449 |
| | 0.45 | 0.7048 | 0.7186 | **0.7340** |
| 1000 | 0.20 | 0.9712 | **0.9833** | 0.9774 |
| | 0.30 | 0.9438 | **0.9619** | 0.9491 |
| | 0.40 | 0.8333 | 0.8537 | **0.8614** |
| | 0.45 | 0.6985 | **0.7308** | 0.7241 |

## E. VLM Output Example

The following is a representative VLM output for a DoH2020 density map with $p = 0.40$ and $N = 500$:

```
[Density map analysis] Dataset: DoH2020 /
    Noise rate: 40% / N=500
CH2 (malicious) high-density region:
    feature [7, 9, 11] / bin [20--25]
CH1 (benign) concentration: feature [0, 1,
    2] / bin [0--5]
Noisy risk region (CH1+CH2 overlap): bin
    [22--24]
Verdict: DoH tunneling pattern -- CH2 high-
    density bin 20--25 of features 7,9,11
RESULT: {"risky_features": [7, 9, 11], "
    risky_bins": [20, 21, 22, 23, 24], "
    reason": "DoH CH2 high-density + CH1
    overlap zone"}
```

The output follows a dual-format design: natural language descriptions for human interpretability and structured JSON for automated parsing. This enables both operational IDS interpretability and seamless integration into programmatic pipelines.

