# OpenReview forum: "VLM-Guided Noisy Label Detection for Structured Network Traffic in Low-Resource IDS"
_ICML.cc/2026/Workshop/FMSD — FMSD @ ICML 2026 Poster_

### Official Review · Reviewer_v8CW · 2026-05-19
**Review of VLM-Guided Noisy Label Detection for Structured Network Traffic in Low-Resource IDS**

**Rating:** 4
**Confidence:** 4

**Review:**

**Summary:**
The paper proposes VLM-IDS, a noisy-label detection pipeline for low-resource intrusion detection. The method first filters noisy network-flow samples using CoTeaching-style XGBoost, converts filtered 50-dimensional packet-length features into three-channel density maps, and then uses a fine-tuned Qwen2-VL-2B model to identify risky feature-bin regions with high benign/malicious overlap. These VLM outputs guide selective augmentation, ACD-based filtering, and feature injection before training a Random Forest classifier. The paper evaluates on DoH2020 and Gotham2025 under synthetic symmetric label noise from 20–45%, reporting consistent gains over raw Random Forest, XGBoost, and RAPIER.

**Strengths:**
1. **The problem is well-motivated:** Low-resource IDS deployments often face noisy labels, scarce annotations, and structured network-flow data where standard supervised learning can degrade significantly.
2. **Interesting representation method:** Shannon entropy density map is an interesting representation. It makes class-overlap regions visually explicit and provides a plausible interface for VLM-based localization.
3. **Experimental grid and results:** The empirical grid is reasonably broad for a workshop paper with two datasets, four noise rates, and multiple random seeds. The results also show clear improvements on the datasets, especially at higher noise rates.
4. **Relevance to workshop:** The work explores a multimodal bridge between structured data and foundation models by converting tabular/time-series flow statistics into density-map images for VLM analysis.

**Areas for Improvement:**
The major limitation in this work is an ablation study to determine attribution. Several of the below areas for improvement stem from this.
1. **Missing ablation study:** The pipeline combines CT filtering, ACD, selective augmentation, feature injection, and VLM guidance, but the paper does not isolate which component drives the gains. It is unclear why VLM reasoning is necessary. Since the entropy map already encodes class overlap explicitly, the paper should compare against simpler alternatives such as heuristic/rule-based methods like direct entropy thresholding, overlap-score rules.
2. **Missing justification for visual representation:** The same density/overlap statistics could be provided as structured text, tabular features, or JSON to a text model or classical classifier. The paper should also compare against non-visual reasoning baselines. This would help justify the image-based VLM formulation.
3. **Why fine-tune a VLM:** The paper does not report an off-the-shelf VLM baseline, so it is unclear whether fine-tuning is essential or whether the base VLM already has enough spatial reasoning ability for this task.
4. **Simple noise formulation:** The noise model is limited to symmetric random label flips. It would be useful to evaluate class-dependent, feature-dependent, or attack-family-dependent noise.

**Justification of Score:**
Overall, the paper offers a promising direction for detecting noisy labels in structured network traffic using VLMs. The problem is relevant, and the proposed entropy-density representation is an interesting way to connect structured data with vision-language models. However, the experiments and intuition do not sufficiently justify the proposed methodology. In particular, it is unclear whether the VLM is genuinely providing useful reasoning, or whether similar gains could be achieved through simpler rule-based methods, text/structured representations, or classical models operating on the same overlap statistics. The lack of ablations makes it difficult to separate the contribution of the VLM from the effects of CT filtering, ACD, augmentation, and feature injection. Therefore, I find the direction interesting, but the current evidence under-validates the main claim.

---

### Official Review · Reviewer_9ntV · 2026-05-20

**Rating:** 4
**Confidence:** 4

**Review:**

The authors propose VLM-IDS, a noisy-label detection and correction pipeline for low-resource intrusion detection. The method first applies a Co-Teaching-style filter with XGBoost learners, converts structured network-flow features into three-channel Shannon Entropy density maps, uses a fine-tuned Qwen2-VL-2B model to identify high-risk feature-bin regions, and then applies adaptive clustering, selective augmentation, and feature injection before training a Random Forest classifier. Experiments are reported on DoH2020 and Gotham2025 with small training sizes and symmetric label noise from 20–45%, showing consistent gains over raw RF, XGBoost, and RAPIER in most settings.

Strengths:
1. Mapping tabular IDS features into entropy-based density images and using a VLM to localize noisy regions is a creative idea. The tabular-to-image bridge is the most novel part of the paper.
2. Low-resource IDS with noisy labels is an important problem, and the paper evaluates small-data regimes with N ∈ {250, 500, 1000} and high symmetric noise rates up to 45%, which is a meaningful stress test.
3. Pipeline is reasonably well motivated. Co-Teaching handles early filtering, entropy maps highlight class overlap, ACD handles local sub-cluster impurity, and VLM guidance is only used when overlap is high. The overlap-adaptive routing is a sensible engineering choice rather than always invoking the VLM.
4. Empirical gains are non-trivial in some settings, with the strongest result on DoH2020, especially N=500, p=0.40, where VLM-IDS reports 0.9023 F1 vs 0.6862 RF and 0.6493 XGBoost. This is a large improvement and suggests the method may be useful in high-overlap/noisy regimes.
5. The VLM produces natural-language explanations plus structured JSON identifying risky features and bins. This is useful for IDS auditability, although the paper does not rigorously validate the explanations.

Areas for Improvement:
1. Ablations are missing. The paper does not isolate the contributions of CT filtering, ACD, VLM guidance, selective augmentation, noisy-score injection, and 150-density features. Since the method is a large pipeline, it is hard to know whether the VLM is actually responsible for the gains.
2. Baselines are weak/incomplete. The main baselines are raw RF, raw XGBoost, and RAPIER. Several standard noisy-label baselines adapted to tabular data are missing, e.g., confident learning, Area Under the Margin Ranking, CINCER, and SimiFeat.
3. The paper reports 10-seed averages but no standard deviations, confidence intervals, or significance tests. This matters because some Gotham2025 differences are small or even negative. VLM-IDS loses or ties in 4/12 conditions, and the gains at p=0.20 are within 1–2 points. This makes the "10/12 wins" claim difficult to justify.
4. No wall-clock or computational cost comparison. Fine-tuning Qwen2-VL on 71k QA pairs for 13 epochs to clean a few hundred labels is computationally expensive. The paper should report a cost comparison vs RAPIER and discuss whether the gains justify it.
5. The injected features may create a train/test mismatch, since noisy scores vary during training but are set to zero at inference, while training-set density statistics are reused for test samples. Since RFs can split on these features, the paper should justify this design and include ablations without noisy-score and density-stat injection.
6. Experimentation with symmetric noise only. Asymmetric/class-conditional noise is more realistic for IDS (adversarial mislabeling is rarely symmetric), and is exactly the regime RAPIER was designed for. The authors claim RAPIER does well under asymmetric noise but never test with it.
7. Overall presentation of the paper has a lot of room for improvement. There are several typos, e.g., "lose" is incorrectly written as "loss", "gegions", "rom", "dimenionality", "seperability", "iming". The conclusion has a paragraph of text from the ICML template boilerplate. Equation for OS is referenced as "OS = Σ min(be_density, ma_density)" but not defined with full notation.

Overall, while the approach is genuinely creative and the empirical gains over RAPIER at high noise are real and meaningful, the paper has a serious attribution problem. With no ablations, I cannot tell whether the VLM is contributing anything beyond what CT + density-feature injection already provide, and the authors' own admission of repetitive VLM outputs makes me believe the VLM is the weakest link in the pipeline. Combined with the symmetric-noise-only evaluation, lack of advanced baselines, missing statistics, and potential train/test mismatch, the paper is not strong enough for a clear acceptance.

---

### Official Review · Reviewer_zdXm · 2026-05-22
**VLM-Guided Noisy Label Detection for Structured Network Traffic in Low-Resource IDS**

**Rating:** 7
**Confidence:** 4

**Review:**

# Summary

VLM-IDS is a multi-stage pipeline for noisy label detection in network intrusion detection under low-data, high-noise conditions. Tabular flow statistics are converted to three-channel Shannon Entropy density maps, which a fine-tuned Qwen2-VL-2B model analyzes to localize high-risk feature regions. This guides selective augmentation and feature injection into a Random Forest classifier. Evaluated on two datasets (DoH2020, Gotham2025) across four noise rates and three training sizes, the system outperforms a raw RF baseline and the prior method RAPIER in most conditions.

# Strengths

- The robustness of VLMs is a major strength when compared to the RAPIER model. RAPIER falls below F1=0.30 at noise rates above 40%, a catastrophic failure for a deployed intrusion detection system. VLM-IDS maintains consistent performance across all tested noise rates, which is the property that actually matters in adversarial real-world settings where noise doesn't have a fixed behavior.
- The conversion of feature-bin density distributions into RGB images where Shannon Entropy highlights class overlap was creative!
- I liked that the authors self-identified issues with VLM-IDS when compared to Gotham2025 on tasks like ACD removal at low noise and 150 injected density features adding noise on already separable data, and offered credible solutions for both.

# Areas of Improvement

- It was unclear to me where in the VLM's pipeline the major contribution comes from. The VLM has five stages: CT filtering, density map construction, VLM routing, ACD sub-clustering, and selective augmentation. It was not clear what the VLM specifically contributes over a pipeline that uses the density map and ACD without VLM guidance. Given the engineering complexity of fine-tuning a vision model on 71,280 QA pairs, this is a significant gap.
- The VLM is fine-tuned on QA pairs derived from ground-truth mislabeling positions. In deployment, ground-truth noise positions are unknown. The paper does not explain how this fine-tuning generalizes to unseen noise patterns, or whether the reported results are partly an artifact of training the VLM on information that implicitly encodes the test conditions.
- Reading a little bit more about RAPIER, it seems like RAPIER's GAN-based architecture is known to be unstable under symmetric noise. Could the authors find better baselines to compare against?

# Recommendation

Good paper, accept. I think the core idea is creative, and the robustness observed at high noise rates seems quite useful in real world deployment. The authors could establish one or two more baselines and explain the VLM's major source of gain a little better. I would especially be interested in the author's take on "Area of improvement point #2."